# Suppress Numerical Oscillations in Transient Mixed Flow Simulations with a Modified HLL Solver

**Zhonghao Mao, Guanghua Guan**  **and Zhonghua Yang** *

State Key Laboratory of Water Resources and Hydropower Engineering Science, Wuhan University, Wuhan 430072, China; 2011301580373@whu.edu.cn (Z.M.); ggh@whu.edu.cn (G.G.)
* Correspondence: yzh@whu.edu.cn

**Abstract:** Transition between free-surface and pressurized flows is a crucial phenomenon in many hydraulic systems. During simulation of such phenomenon, severe numerical oscillations may appear behind filling-bores, causing unphysical pressure variations and computation failure. This paper reviews existing oscillation-suppressing methods, while only one of them can obtain a stable result under a realistic acoustic wave speed. We derive a new oscillation-suppressing method with first-order accuracy. This simple method contains two parameters, $P_a$ and $P_b$, and their values can be determined easily. It can sufficiently suppress numerical oscillations under an acoustic wave speed of 1000 ms$^{-1}$. Good agreement is found between simulation results and analytical results or experimental data. This paper can help readers to choose an appropriate oscillation-suppressing method for numerical simulations of flow regime transition under a realistic acoustic wave speed.

**Keywords:** flow regime transition; finite volume methods; numerical oscillations; numerical viscosity; Preissmann slot model

---

## 1. Introduction

In water conveyance systems, water flows under free-surface flow condition or pressurized flow condition. Under certain circumstances, transition between the two flow regimes may occur (i.e., the flow regime transition phenomenon). Following the transition, force exerting on structures changes violently and causes structural damage [1–3]. Numerical simulation of flow regime transition can provide substantial information for the design and management of river-crossing bridges, tunnels, conducts and culverts [4–10].

The complexity of flow regime transition lies in the presence of free-surface and pressurized flows, which are governed by different equations. This problem can be avoided by adopting one set of governing equations for two flow regimes. Based on this idea, the Preissmann slot model (PSM) is proposed [11]; it was adopted by many researchers and commercial software packages [12–18]. The strong gradient in piezometric head at the interface between two flow regimes forms a discontinuity in flow. Finite volume methods can capture the discontinuity in the flow implicitly, which makes them popular in simulating flow regime transition [19].

Despite of all the fine properties that finite volume methods have, the numerical oscillations in a flow regime transition simulation have troubled many hydraulic engineers [12]. These numerical oscillations have the same origin with "post-shock oscillations" in gas dynamics [20–23]. In analytical results, the thickness of the filling-bore is infinitely small, and the flow states at the two sides of a filling-bore satisfy the Rankine–Hugoniot condition. In numerical simulations, a filling-bore spreads over several computational cells, and the flow states at the two adjacent cells do not satisfy the Rankine–Hugoniot condition. This causes trivial discrepancies in the mass and momentum fluxes, which are amplified in simulation results due to the large acoustic wave speed. High-order finite volume

methods cause more numerical oscillations because of low dissipation away from the shocks [21]. First-order upwind finite volume methods fail to prevent numerical oscillations without compromising the representation of the filling-bore [23–25].

A lot of effort has been spent to obtain a stable and accurate result of flow regime transition [12,23,24]. In this paper, some existing oscillation-suppressing methods are tested on a benchmark model. Considering the lack of an efficient and simple method, the authors derive a new method, which can suppress numerical oscillations and capture the filling-bore nicely under a high acoustic wave speed. The structure of this paper is as follows: Section 2 introduces the governing equations and the discretization method. Section 3 reviews the existing oscillation-suppressing methods, and their effects are evaluated on the benchmark model that was adopted by Malekpour and Karney [24]. Section 4 proposes a new and simple modified HLL solver to suppress numerical oscillations. Its accuracy and robustness are tested against the analytical results and experiment data in Section 5. Conclusions are drawn in the last section.

## 2. Governing Equations and Discretization Method

The PSM places an infinitely high narrow slot on the top of the conduct, so that it becomes an open-channel with a composite cross-section. The water depth in the slot represents the piezometric head of the pressurized flow inside the original conduct, as shown in Figure 1. The slot width needs to be very small so that the gravity wave speed inside it is identical to the acoustic wave speed.

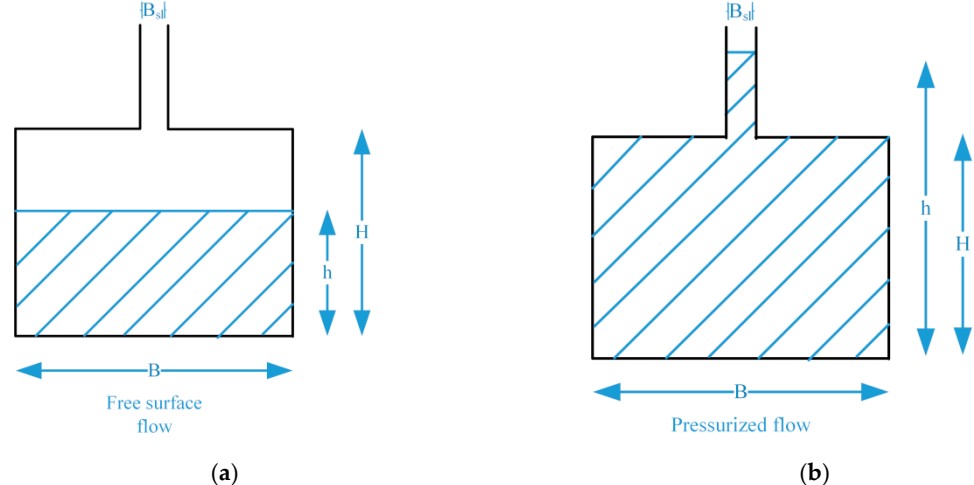

**(a)**　　　　　**(b)**

**Figure 1.** A rectangular conduct with a slot on its top: (**a**) free surface flow; (**b**) pressurized flow.

Under the framework of PSM, the governing equations of one-dimensional flow regime transition with uniform cross-sections can be written as [26]:

$$\frac{\partial \mathbf{U}}{\partial t} + \frac{\partial \mathbf{F}(\mathbf{U})}{\partial x} = \mathbf{S}(\mathbf{U})$$

$$\mathbf{U} = \begin{pmatrix} A \\ Q \end{pmatrix}, \mathbf{F}(\mathbf{U}) = \begin{pmatrix} Q \\ \frac{Q^2}{A} + gI(h) \end{pmatrix}, \mathbf{S}(\mathbf{U}) = \begin{pmatrix} 0 \\ gA(S_b - S_f) \end{pmatrix} \tag{1}$$

$$I(h) = \int_0^h (h - \xi) l(x, \xi) d\xi$$

where $Q$ is the volume flow rate, $A$ is the wetted area, $g$ is the acceleration of gravity, $h$ is the water depth, $I$ is the moment of inertia, $l$ is the cross-sectional width and $l(x, h) = b(x)$ the water surface width.

The external force acting on the flow is accounted in the source term, where $S_b$ is the bed slope and $S_f$ is the friction slope, which can be computed using the Manning relation:

$$S_f = \frac{n^2 u |u|}{R^{4/3}} \tag{2}$$

where $u$ is the flow velocity, $n$ is the Manning coefficient and $R$ is the hydraulic radius. For a rectangular cross-section with a slot on its top, the parameters $b$, $A$, $I$ and wave speed $c$ can be expressed as functions of $h$:

$$b(h) = \begin{cases} B & h \leq H \\ B_{sl} & h > H \end{cases} \tag{3}$$

$$A(h) = \begin{cases} Bh & h \leq H \\ BH + B_{sl}(h - H) & h > H \end{cases} \tag{4}$$

$$I(h) = \begin{cases} 0.5Bh^2 & h \leq H \\ BH(h - 0.5H) + 0.5B_{sl}(h - H)^2 & h > H \end{cases} \tag{5}$$

$$c(h) = \begin{cases} \sqrt{gh} & h \leq H \\ \sqrt{g\frac{A}{B_{sl}}} & h > H \end{cases} \tag{6}$$

where $H$ and $B$ are the cross-sectional height and width, and $B_{sl}$ is slot width. In order to make the gravity wave speed inside the slot equal to the acoustic wave speed $a$, the slot width $B_{sl} = gA_f\,a^{-2}$, where $A_f$ is the full cross-sectional area of the conduct [27]. Using the Godunov-type finite volume methods with first-order accuracy and assuming a piecewise constant data construction, the governing equations are discretized as

$$\mathbf{U}_i^* = \mathbf{U}_i^n - \frac{\Delta t_i}{\Delta x_i}\left(\mathbf{F}_{i+1/2} - \mathbf{F}_{i-1/2}\right) \tag{7}$$

$$\mathbf{U}_i^{n+1} = \mathbf{U}_i^* + \Delta t_i \mathbf{S}\left(\mathbf{U}_i^*\right) \tag{8}$$

## 3. Review of Current Oscillation-Suppressing Methods

The benchmark model was proposed by Malekpour and Karney [24], it consists of a conduct with square-unit cross-sections that are connected to a reservoir at the upstream end. The acoustic wave speed is 1000 ms$^{-1}$ and the slot width is $9.8 \times 10^{-6}$ m. Under the initial condition, 0.6 m-deep stagnant water is in the conduct while the water level inside the reservoir is constantly 4 m, as shown in Figure 2.

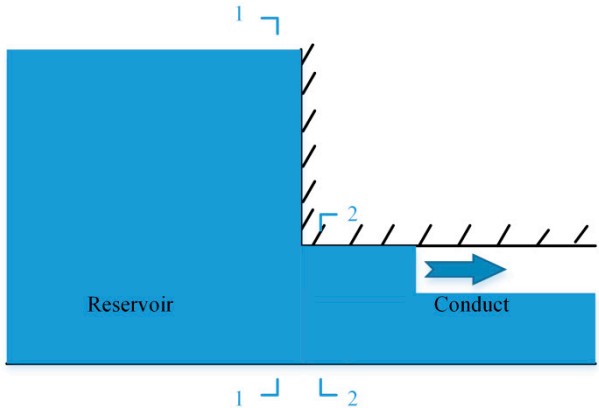

**Figure 2.** Front view of the benchmark model.

Under initial condition, flow states $\mathbf{U}_L$ in the reservoir and $\mathbf{U}_R$ in the conduct are discontinuous:

$$\begin{bmatrix} h_L \\ u_L \end{bmatrix} = \begin{bmatrix} 4 \text{ m} \\ 0 \text{ ms}^{-1} \end{bmatrix}, \begin{bmatrix} h_R \\ u_R \end{bmatrix} = \begin{bmatrix} 0.6 \text{ m} \\ 0 \text{ ms}^{-1} \end{bmatrix} \tag{9}$$

Since $h_L$ is larger than $h_R$, a shock wave (filling-bore) belonging to the second characteristic field is formed at the conduct inlet and propagates downstream. Consider 1–1 as a cross-section in the reservoir and 2–2 as a cross-section at conduct inlet: flow sates at 1–1 and 2–2 are $\mathbf{U}_1$ and $\mathbf{U}_2$, respectively. Then $\mathbf{U}_2$ can be obtained by solving the following equations iteratively [24]:

$$h_1 = h_2 + \frac{u_2^2}{2g} \tag{10}$$

$$u_2 = u_R + \sqrt{\frac{[gI(A_R) - gI(A_2)](A_R - A_2)}{A_R A_2}} \tag{11}$$

where $\mathbf{U}_1 = \mathbf{U}_L$ by assuming that flow velocity inside the reservoir is negligible. In this case, $h_2$ and $u_2$ are 3.167 m and 4.0334 ms$^{-1}$, a ghost cell is set at the upstream boundary adopting $h_2$ and $u_2$. Since $\mathbf{U}_2$ is connected to $\mathbf{U}_R$ through a right shock, the flow states inside the conduct ultimately will take place by $\mathbf{U}_2$. The propagation speed of the filling-bore is given by the Rankine–Hugoniot condition, which is 10.067 ms$^{-1}$. Then we can construct the analytical result in the benchmark model at $t_0$:

$$h(x) = \begin{cases} 3.167 \text{ m} & x < 10.067t_0 \\ 0.6 \text{ m} & x > 10.067t_0 \end{cases}, u(x) = \begin{cases} 4.0334 \text{ ms}^{-1} & x < 10.067t_0 \\ 0 \text{ ms}^{-1} & x > 10.067t_0 \end{cases} \tag{12}$$

As customary, $x$ denotes the distance to the conduct inlet. In a numerical simulation, the size of each computational cell is 1 m, the time step is 0.008 s and the Courant number is 0.8. When the acoustic wave speed adopted in the simulation exceeds 100 ms$^{-1}$, the magnitude of the numerical oscillations become so large that the simulated piezometric head become negative, and the simulation will not proceed [24]. In the remaining part of this section, the readers will see that only one method can get a satisfactory result under a high acoustic wave speed, while its performance rely on two parameters which must be well tuned. This shows the importance of devising an alternative method, which is stable and convenient.

### 3.1. Numerical Filtering Method

In this method, the exact Riemann solver is adopted to solve the Riemann problem at each cell boundary. Flow states $\mathbf{U}_{i+1/2}$ at $x_{i+1/2}$ satisfy the following equations:

$$u_{i+1/2} = \begin{cases} u_i - \sqrt{g\frac{[I(A_i) - I(A_{i+1/2})](A_i - A_{i+1/2})}{A_i A_{i+1/2}}} & A_{i+1/2} > A_i \\ u_i + \int_0^{A_L}\sqrt{\frac{g}{\alpha b}}d\alpha - \int_0^{A_{i+1/2}}\sqrt{\frac{g}{\alpha b}}d\alpha & A_{i+1/2} < A_i \end{cases} \tag{13}$$

$$u_{i+1/2} = \begin{cases} u_{i+1} + \sqrt{g\frac{[I(A_{i+1}) - I(A_{i+1/2})](A_{i+1} - A_{i+1/2})}{A_{i+1} A_{i+1/2}}} & A_{i+1/2} > A_{i+1} \\ u_{i+1} - \int_0^{A_{i+1}}\sqrt{\frac{g}{\alpha b}}d\alpha + \int_0^{A_{i+1/2}}\sqrt{\frac{g}{\alpha b}}d\alpha & A_{i+1/2} < A_{i+1} \end{cases} \tag{14}$$

Equations (13) and (14) can be solved iteratively, then the wave structure in the Riemann problem can be determined and utilized to compute the flux; see Kerger et al. [26] for detail. Although the exact solver can obtain the complete wave structure, serious numerical oscillations appear in the simulation

result. Vasconcelos et al. [23] proposed to suppress the numerical oscillations by averaging the flow states among the three conjunct cells at each time step:

$$\mathbf{U}_i^{n+1} = (1 - 2\varepsilon)\mathbf{U}_i^{n+1} + \varepsilon\left(\mathbf{U}_{i-1}^{n+1} + \mathbf{U}_{i+1}^{n+1}\right) \tag{15}$$

The authors suggest $\varepsilon$ to be between 0.025 and 0.050. This method will increase the spreading length of the filling-bore front and remove any physical oscillations that appear in the solution. The simulation result of this method using $\varepsilon = 0.04$ is drawn in Figure 3.

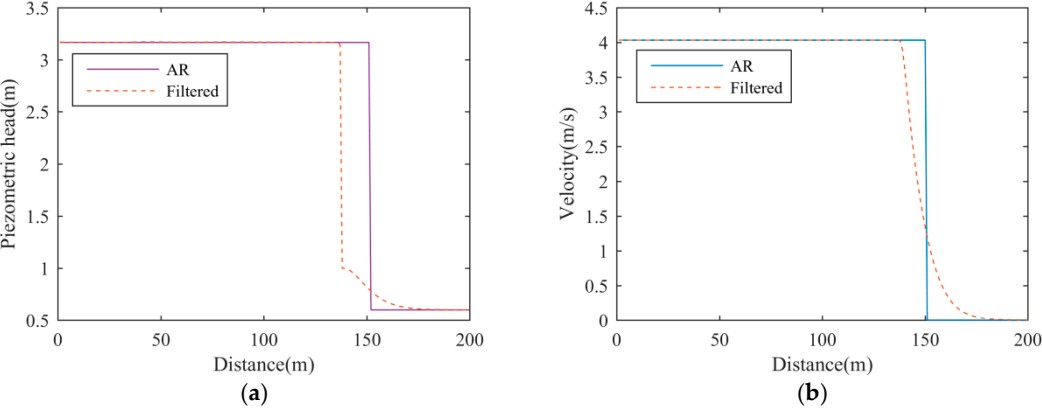

**Figure 3.** Comparison of results simulated by the numerical filtering method (filtered) and the analytical result (AR): (**a**) piezometric head; (**b**) flow velocity.

*3.2. Hybrid Flux Method*

The hybrid flux method uses two types of numerical fluxes alternatively. The first one is based on the Roe solver [28] and the LxF scheme [23]. Define $\lambda_{i+1/2}^{j}$ as eigenvalues of the linearized Jacobian matrix, $\mathbf{R}_{i+1/2}^{j}$ as the corresponding right eigenvectors and $\alpha_{i+1/2}^{j}$ as the wave strengths across the cell boundary:

$$\lambda_{i+1/2}^{1} = \frac{\widetilde{Q}_{i+1/2}}{\widetilde{A}_{i+1/2}} - \widetilde{c}_{i+1/2}, \lambda_{i+1/2}^{2} = \frac{\widetilde{Q}_{i+1/2}}{\widetilde{A}_{i+1/2}} + \widetilde{c}_{i+1/2} \tag{16}$$

$$\mathbf{R}_{i+1/2}^{1} = \left[1, \lambda_{i+1/2}^{1}\right]^{T}, \mathbf{R}_{i+1/2}^{2} = \left[1, \lambda_{i+1/2}^{2}\right]^{T} \tag{17}$$

$$\alpha_{i+1/2}^{1} = \frac{\lambda_{i+1/2}^{2}(A_{i+1} - A_i) - (Q_{i+1} - Q_i)}{\lambda_{i+1/2}^{2} - \lambda_{i+1/2}^{1}}, \alpha_{i+1/2}^{2} = \frac{(Q_{i+1} - Q_i) - \lambda_{i+1/2}^{1}(A_{i+1} - A_i)}{\lambda_{i+1/2}^{2} - \lambda_{i+1/2}^{1}} \tag{18}$$

where $\widetilde{Q}_{i+1/2}$, $\widetilde{A}_{i+1/2}$ and $\widetilde{c}_{i+1/2}$ are Roe averages:

$$\widetilde{A}_{i+1/2} = (A_i A_{i+1})^{1/2} \tag{19}$$

$$\widetilde{Q}_{i+1/2} = \frac{Q_{i+1}(A_i)^{1/2} + Q_i(A_{i+1})^{1/2}}{(A_i)^{1/2} + (A_{i+1})^{1/2}} \tag{20}$$

$$\widetilde{c}_{i+1/2} = \begin{cases} \left[g\frac{I(A_{i+1}) - I(A_i)}{A_{i+1} - A_i}\right]^{1/2} & A_{i+1} \neq A_i \\ \left[g\frac{A_i + A_{i+1}}{b_i + b_{i+1}}\right]^{1/2} & A_{i+1} = A_i \end{cases} \tag{21}$$

Then the fluxes obtained by the Roe solver are written as

$$\mathbf{F}_{i+1/2}^{\mathrm{Roe}} = \frac{1}{2}\left[F\left(\mathbf{U}_{i+1}^n\right) + F\left(\mathbf{U}_i^n\right)\right] - \frac{1}{2}\sum_{j=1}^{2}\left|\lambda_{i+1/2}^j\right|\alpha_{i+1/2}^j\mathbf{R}_{i+1/2}^j \tag{22}$$

The Roe solver is known to be vulnerable to numerical oscillations [21,29], while the LxF scheme is robust against numerical oscillations but causes too much diffusion of the filling-bore:

$$\mathbf{F}_{i+1/2}^{\mathrm{LxF}} = \frac{1}{2}\left[F\left(\mathbf{U}_{i+1}^n\right) + F\left(\mathbf{U}_i^n\right)\right] - \frac{\Delta x}{2\Delta t}(\mathbf{U}_{i+1} - \mathbf{U}_i) \tag{23}$$

Compare Equations (22) with (23): The difference between the Roe solver and LxF scheme is the choice of eigenvalues. Vasconcelos et al. [23] proposed a new method to determine the eigenvalues:

$$\left|\lambda_{i+1/2}^{1,2}\right| = \min\left[\frac{\Delta x}{\Delta t}, \left|\frac{\widetilde{Q}_{i+1/2}}{\widetilde{A}_{i+1/2}} \mp \widetilde{c}_{i+1/2}\right| + L(\Delta c_{i+1/2})^L\frac{\Delta x}{\Delta t}\right] \tag{24}$$

$$\Delta c_{i+1/2} = \frac{\left|\widetilde{c}_{i+1} - \widetilde{c}_i\right|}{\max\left(\left|\widetilde{c}_{i+1} - \widetilde{c}_i\right|\right)_{i=1\dots N-1}} \tag{25}$$

This method is referred to as the hybrid one from here on further. At the cell boundary between the free-surface and pressurized flows, $\Delta c_{i+1/2} = 1$, as $L$ changes from 0 to 1, the eigenvalues switches from those obtained by the Roe solver to those adopted in the LxF scheme. At the other cell boundaries, $\Delta c_{i+1/2}$ is approximately 0, thus the eigenvalues in Equation (24) remain close to those obtained by the Roe solver. In this way, numerical viscosity is added at the cell boundary where flow condition transition happens, and its amount increases with $L$. The simulation results of the hybrid one with $L = 0.6$ are drawn in Figure 4. This method overestimates the spreading length of the filling-bore, and it fails to suppress the numerical oscillations under a high acoustic wave speed.

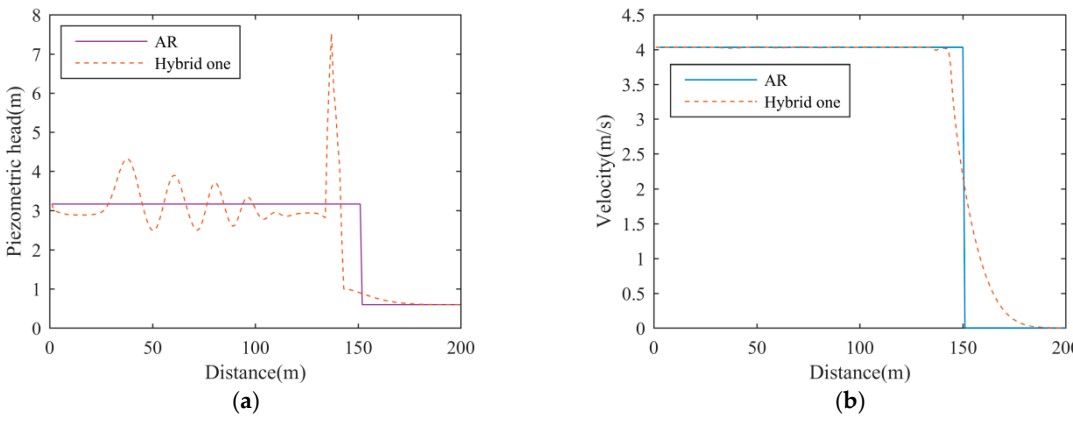

**Figure 4.** Comparison of the results simulated by hybrid one and the analytical result (AR): (**a**) piezometric head; (**b**) flow velocity.

Hyunuk et al. [12] chose different fluxes; they used the FORCE scheme [30] at the cell boundary between the free-surface and pressurized flows, and the HLL solver [30] elsewhere:

$$\mathbf{F}_{i+1/2}^{\mathrm{hybrid}} = \begin{cases} \mathbf{F}_{i+1/2}^{\mathrm{FORCE}} & (h_L - H_L)(h_R - H_R) < 0 \\ \mathbf{F}_{i+1/2}^{\mathrm{HLL}} & \text{otherwise} \end{cases} \tag{26}$$

This method is referred to as hybrid two from here on further. The HLL solver starts by estimating the largest wave speed $S_{wR}$ and smallest wave speed $S_{wL}$ in the Riemann problem. The fluxes are computed based on the signs of $S_{wL}$ and $S_{wR}$:

$$\mathbf{F}_{i+1/2}^{HLL} = \begin{cases} F(\mathbf{U}_i) & S_{wL} > 0 \\ \frac{S_{wR}F(\mathbf{U}_i) - S_{wL}F(\mathbf{U}_{i+1}) + S_{wR}S_{wL}(\mathbf{U}_{i+1} - \mathbf{U}_i)}{S_{wR} - S_{wL}} & S_{wL} \leq 0 \quad \text{and} \quad S_{wR} \geq 0 \\ F(\mathbf{U}_{i+1}) & S_{wR} < 0 \end{cases} \tag{27}$$

The choices of $S_{wL}$ and $S_{wR}$ follow Toro [31]:

$$S_{wL} = u_i - \Omega_i, S_{wR} = u_{i+1} + \Omega_{i+1} \tag{28}$$

$$\Omega_{K(K = i,i+1)} = \begin{cases} \sqrt{g\frac{[I(A_{i+1/2}) - I(A_K)]A_{i+1/2}}{A_K(A_{i+1/2} - A_K)}} & A_{i+1/2} > A_K \\ c_K & A_{i+1/2} \leq A_K \end{cases} \tag{29}$$

where $A_{i+1/2}$ is an estimate of the wetted area at $x_{i+1/2}$; we adopt the one proposed by Leno et al. [32], which admits the minimum amount of numerical viscosity:

$$A_{i+1/2} = \frac{A_i + A_{i+1}}{2}\left(1 + \frac{u_i - u_{i+1}}{c_i + c_{i+1}}\right) \tag{30}$$

The FORCE flux can be written as the algebraic average of the LxF scheme and Lax–Wendorff scheme:

$$\mathbf{F}_{i+1/2}^{LW} = F\left(\mathbf{U}_{i+1/2}^{LW}\right), \mathbf{U}_{i+1/2}^{LW} = \frac{1}{2}\left(\mathbf{U}_i^n + \mathbf{U}_{i+1}^n\right) - \frac{\Delta x}{2\Delta t}\left[F\left(\mathbf{U}_{i+1}^n\right) - F\left(\mathbf{U}_i^n\right)\right] \tag{31}$$

$$\mathbf{F}_{i+1/2}^{FORCE} = \frac{1}{2}\left(\mathbf{F}_{i+1/2}^{LW} + \mathbf{F}_{i+1/2}^{LxF}\right) \tag{32}$$

The FORCE scheme is a centred scheme; thus, it is robust against numerical oscillations. It is less diffusive than the LxF scheme, which can reduce the over-smearing at strong gradients [33]. The simulation results of hybrid two are depicted in Figure 5.

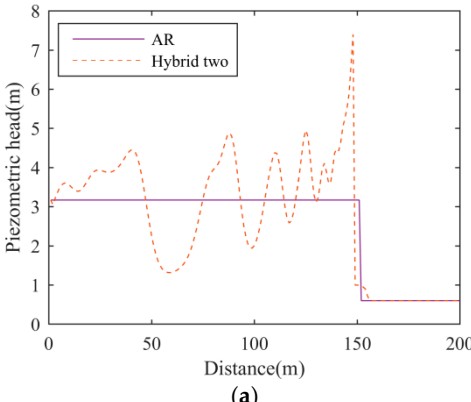 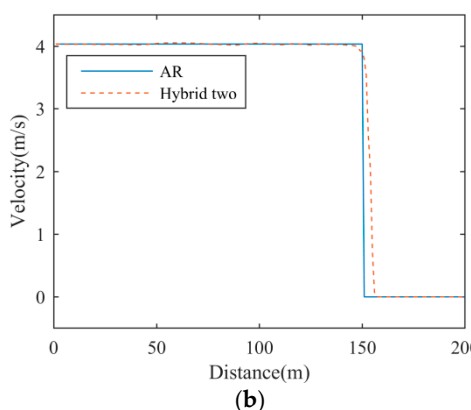

(**a**)　　　　　　　　　　　　　　　　　　　　　　　　　　(**b**)

**Figure 5.** Comparison of the results simulated by hybrid two and the analytical result (AR): (**a**) piezometric head; (**b**) flow velocity.

The hybrid flux methods have added enough numerical viscosity at the cell boundary between the two flow regimes but, still, serious numerical oscillations appear in the simulation results. The reason is that the numerical oscillations appear as soon as the flow regime transition happens, and while the two methods start to add numerical viscosity one time-step falls behind of it. If one method can foresee the happening of the flow regime transition, and admits numerical viscosity ahead of it, it will achieve a more stable result.

### 3.3. Modified HLL Solver

Malekpour and Karney [24] pointed out that the amount of numerical viscosity in the HLL fluxes increases with the magnitude of $S_{wL}$ and $S_{wR}$. In fact, the HLL fluxes equal the LxF fluxes when $|S_{wL}|$ and $|S_{wR}|$ equal $\Delta x / \Delta t$. To increase the amount of numerical viscosity, they proposed a modified HLL solver (referred to as the M_HLL solver). In the M_HLL solver, $A_{i+1/2}$ in Equation (29) is computed according to a reference depth $h_G$:

$$A_{i+1/2} = A(h_G), h_G = K_a \max(h_{i-NS}, h_{i-NS+1}, \cdots, h_{i-1}, h_i, h_{i+1}, \cdots, h_{i+NS-1}, h_{i+NS}) \tag{33}$$

The depth $h_G$ is defined as $K_a$, multiplying the highest piezometric head in the $2NS+1$ cells surrounding cell $i$, while $K_a > 1$ and $NS \geq 3$. The solution of Equation (33) produces a larger magnitude of $S_{wL}$ and $S_{wR}$; thus, increasing the numerical viscosity before the flow regime transition happens. The simulation results of M_HLL with $K_a = 1.4$ and $NS = 5$ are drawn in Figure 6.

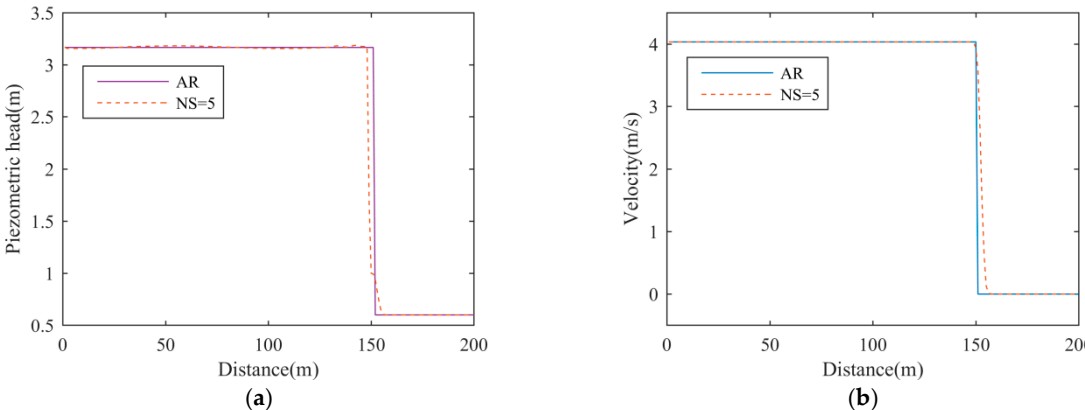

**Figure 6.** Comparison of the results simulated by M_HLL, with $K_a = 1.4$ and $NS = 5$, and the analytical result (AR): (**a**) piezometric head; (**b**) flow velocity.

The M_HLL solver can suppress numerical oscillations in the benchmark model, and it only slightly increases the spreading length at the filling-bore front. However, the values of $K_a$ and $NS$ can affect the diffusion and accuracy of the M_HLL solver to a great extent; see Figure 7.

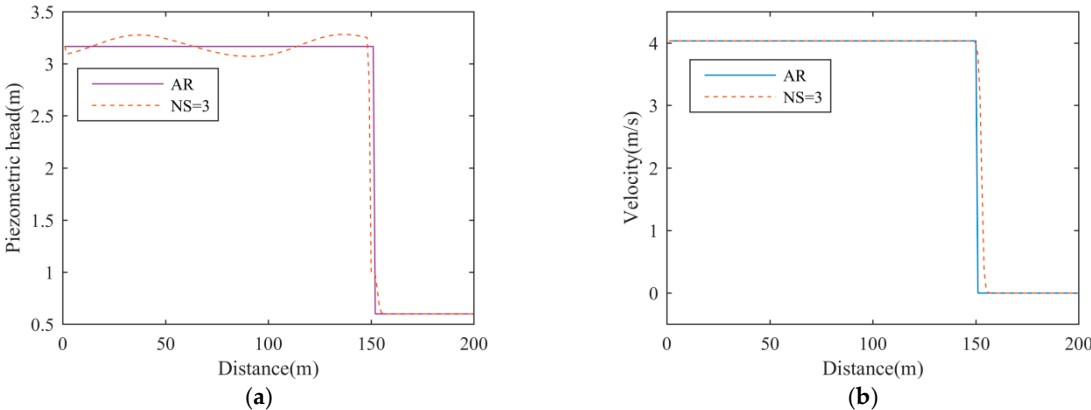

**Figure 7.** Comparison of the results simulated by M_HLL, with $K_a = 1.4$ and $NS = 3$, and the analytical result (AR): (**a**) piezometric head; (**b**) flow velocity.

Thus, the values of $K_a$ and $NS$ must be well-tuned. Meanwhile, the way to determine $h_G$ makes the HLL solver hard to use in parallelized computation. In the next section, the authors present an alternative method, which is equally efficient as the M_HLL solver.

## 4. A New Modified HLL Solver

In this section, we present a new modified HLL solver (referred to as the P_HLL solver) to suppress numerical oscillations. In the P_HLL solver, the solution to $A_{i+1/2}$ depends on the water depths at cell $i$ and $i + 1$. When $h_i$ and $h_{i+1}$ are below $P_bH$, $A_{i+1/2}$ is computed using Equation (30) to admit the minimum amount of numerical viscosity, otherwise $A_{i+1/2}$ is computed according to a constant depth $P_aH$:

$$A_{i+1/2} = A(P_aH), \quad h_i > P_bH \quad \text{or} \quad h_{i+1} > P_bH \tag{34}$$

$P_b$ must be smaller than one, and a preferable value is between 0.6 and 0.8. $P_aH$ must be larger than the piezometric head peak during the transition to admit enough numerical viscosity.

To illustrate the effect of $P_a$ and $P_b$, we study the Riemann problem at $x_{i+1/2}$ in the benchmark model. Suppose $h_i$ and $h_{i+1}$ is 3.167 m and 0.6 m, respectively, and $u_i$ and $u_{i+1}$ is 4.0334 ms$^{-1}$ and 0 ms$^{-1}$, respectively. The solution of Equation (30) lies between $A_i$ and $A_{i+1}$, and after substituting it into Equation (28), one will get $S_{wL}$ (noted as $S_{wL1}$) as the speed of the left rarefaction wave head, and $S_{wR}$ (noted as $S_{wR1}$) as the speed of right shock wave:

$$S_{wL1} = u_i - c_i, S_{wR1} = u_{i+1} + \sqrt{g\frac{[I(A_{i+1/2}) - I(A_{i+1})]A_{i+1/2}}{A_{i+1}(A_{i+1/2} - A_{i+1})}} \tag{35}$$

A sketch of two waves in the Riemann problem is shown in Figure 8.

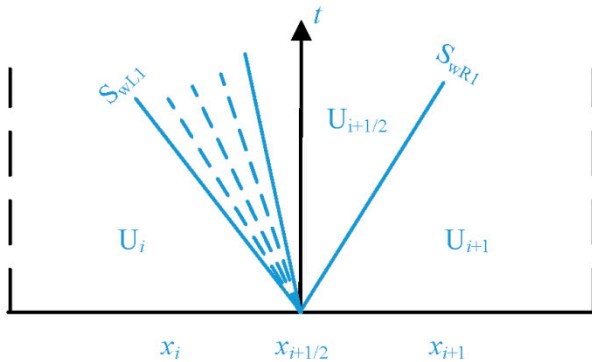

**Figure 8.** A left rarefaction wave with a right shock wave at $x_{i+1/2}$.

Because cell $i$ is under a pressurized flow condition, it is easy to see that $|S_{wL1}|$ nearly equals the acoustic wave speed. The entropy condition of right shock wave requires

$$u_{i+1/2} + c_{i+1/2} > S_{wR1} > u_{i+1} + c_{i+1} \tag{36}$$

The magnitude of $S_{wR1}$ equals the propagation speed of the filling-bore, which is 10.067 ms$^{-1}$, and it is much smaller than the acoustic wave speed.

The solution of Equation (34) is unconditionally larger than $A_i$ and $A_{i+1}$, which produces two shock waves in the Riemann problem. Consequently, $S_{wR2}$ is the speed of the right shock wave, and $S_{wL2}$ is the speed of the left shock wave; see Figure 9.

The entropy condition of the left shock wave requires

$$u_{i+1/2} - c_{i+1/2} < S_{wL2} < u_i - c_i \tag{37}$$

Since cell $i$ is under a pressurized flow condition, $u_i << c_i$; thus, $|S_{wL2}| > |S_{wL1}|$ and they are both close to the acoustic wave speed. The speed of the right shock wave increases with $A_{i+1/2}$; thus, $S_{wR2} > S_{wR1}$. This larger magnitude of $S_{wR}$ admits more mass and momentum into cell $i + 1$ before it becomes pressurized. The loci of $\mathbf{U}_{i+1}$ simulated by HLL and P_HLL ($P_a = 5$, $P_b = 0.7$) are drawn in Figure 10.

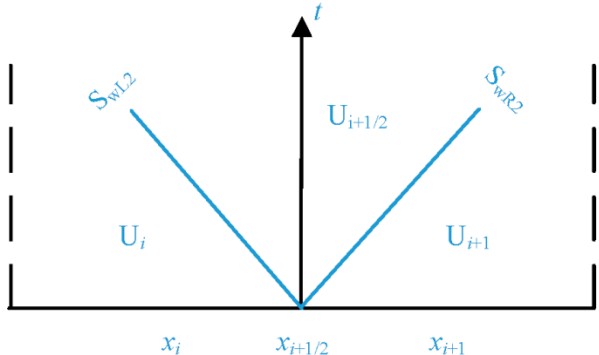

**Figure 9.** A left shock wave with a right shock wave at $x_{i+1/2}$.

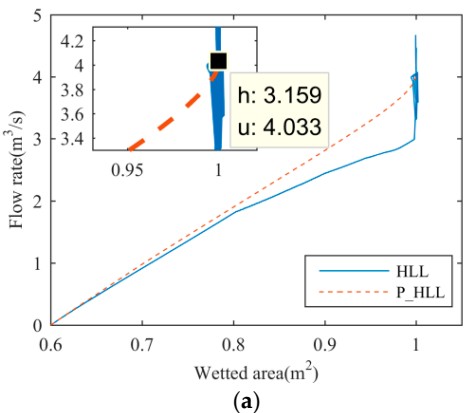

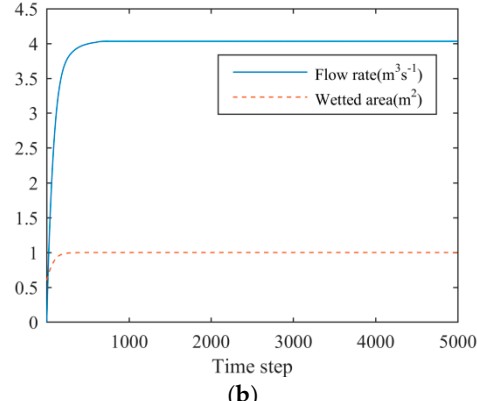

(**a**)

(**b**)

**Figure 10.** (**a**) Locus of the flow states in cell $i + 1$ simulated by HLL and P_HLL ($P_a = 5$, $P_b = 0.7$); (**b**) history of the flow states in cell $i + 1$ simulated by P_HLL ($P_a = 5$, $P_b = 0.7$).

Vertexes appear in the locus simulated by HLL after cell $i + 1$ becomes pressurized, which represent numerical oscillations in the simulation result. In contrast, P_HLL achieves a smooth and stable transition between the free-surface flow and pressurized flow. The locus simulated by P_HLL converges to a 3.159 m depth and 4.033 ms$^{-1}$ velocity, which is very close to the flow states at the entrance of the conduct. The discrepancy in water depth is more pronounced due to the small value of the slot width. Therefore, P_HLL preserves the conservation in mass and momentum.

A larger magnitude of $S_{wR2}$ admits more mass and momentum into cell $i + 1$ before it is pressurized; thus, it increases the diffusion of the filling-bore. The magnitude of $S_{wL2}$ and $S_{wR2}$ are related to the value of $A_{i+1/2}$, which consequently depends on the value of $P_a$; see Figures 11 and 12.

Figure 11 shows that $S_{wR2}$ increases with $P_a$, while $|S_{wL2}|$ barely changes with $P_a$ and stays close to the acoustic wave speed. However, $S_{wR2}$ does not increase significantly when $P_a$ changes from 1 to 10, which denotes that the diffusion of the filling-bore does not increase significantly when $P_a$ changes from 1 to 10. The simulation results using the P_HLL solver with $P_b = 0.7$ and different values of $P_a$ are drawn in Figure 13.

Although $P_a = 10$ produces a more diffusive filling-bore, the spreading length of the filling-bore does not increase significantly compared to that using $P_a = 5$. During a realistic transition phenomenon, the piezometric head peak seldom exceeds 10 times the cross-sectional depth. Therefore, one can always start by adopting a large $P_a$ (for example 10) in the P_HLL solver and do not worry about significantly compromising the representation of the filling-bore.

Compared to $P_a$, the value of $P_b$ has a more significant effect on the numerical oscillations, for it determines the threshold depth where numerical viscosity starts to increase. $P_b$ must be smaller than one so that the numerical viscosity is added before the flow regime transition happens. A smaller $P_b$

leads to more stable result, but it may cause more diffusion. The simulation results using $P_a = 5$ and $P_b = 0.7$ or $0.9$ are drawn in Figure 14.

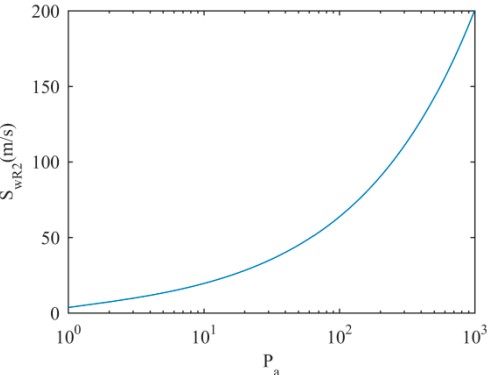

**Figure 11.** $S_{wR2}$ under different values of $P_a$.

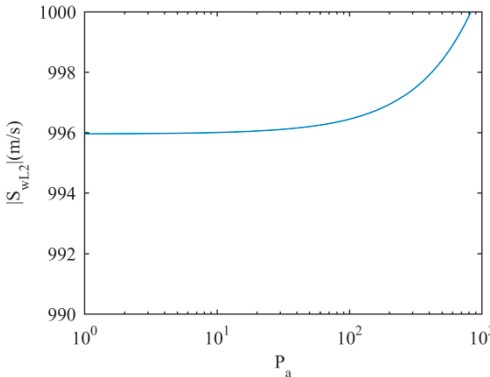

**Figure 12.** $|S_{wL2}|$ under different values of $P_a$.

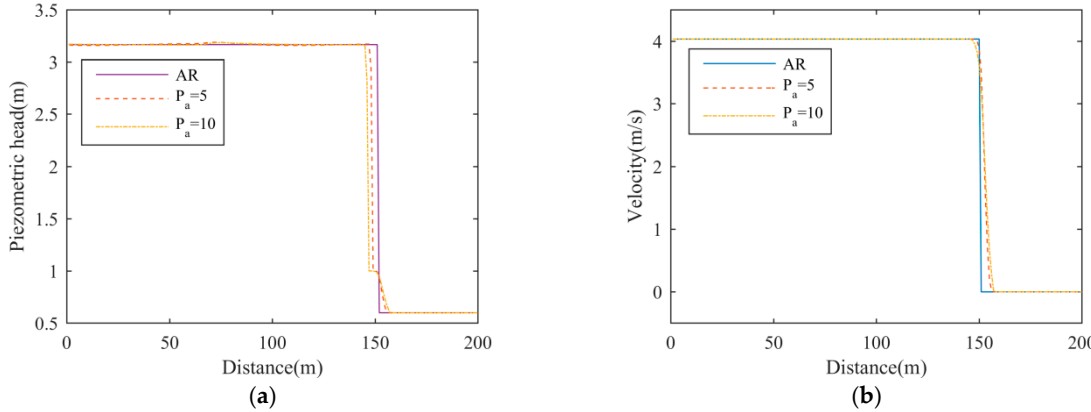

**Figure 13.** Comparison of the results simulated by P_HLL, with $P_b = 0.7$ and $P_a = 5$ or $P_a = 10$, and the analytical result (AR): (**a**) piezometric head; (**b**) flow velocity.

When $P_b = 0.9$, a smooth transition between the free-surface and pressurized flows cannot be guaranteed. Therefore, we suggest a $P_b$ between 0.6 and 0.8 to avoid causing two much diffusion of the filling-bore. This is also supported by the numerical tests in the next section.

In conclusion, at a free-surface cell, P_HLL admits numerical viscosity once the water depth is higher than a threshold value $P_b H$. Thus, a smooth transition from the free-surface flow to pressurized flow can be obtained. Meanwhile, P_HLL causes diffusion of the filling-bore. In realistic applications,

a $P_a$ of 10 is large enough to suppress numerical oscillations without significantly increasing the spreading-length of the filling-bore. The value of $P_b$ is suggested to be between 0.6 and 0.8.

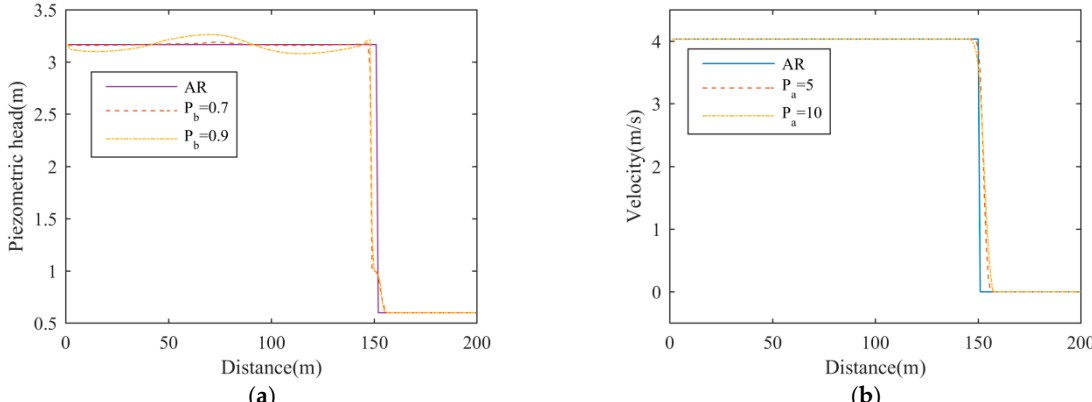

(a)                                                               (b)

**Figure 14.** Comparison of the results simulated by P_HLL, with $P_a$ = 5 and $P_b$ = 0.7 or $P_b$ = 0.9, and the analytical result (AR): (**a**) piezometric head; (**b**) flow velocity.

## 5. Numerical Tests

### 5.1. Two Filling-Bores

This test evaluates the accuracy of P_HLL under the presence of two filling-bores. It consists of a 200 m-long conduct with square-unit cross-sections and two reservoirs connected to it at the upstream end and downstream end; the acoustic wave speed is 1000 ms$^{-1}$. At initial conditions, water in the conduct is stationary with a depth of 0.6 m, while water depth at the upstream and downstream reservoir is 4 m and 3 m, respectively. The model set up is sketched in Figure 15.

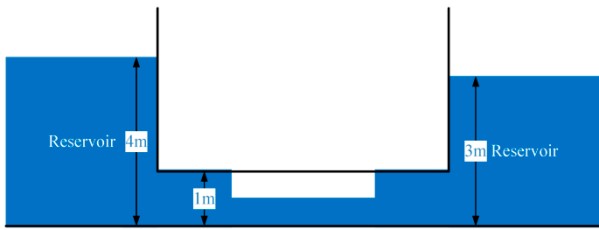

**Figure 15.** A sketch of the reservoir–conduct–reservoir model.

At $t$ = 0 s, the conduct inlet and outlet are opened immediately, forming two filling-bores that propagate in the opposite direction. The upstream filling-bore is identical with that in the benchmark model. The downstream filling-bore can be derived analogously; its propagation speed is 8.429 ms$^{-1}$, and the flow states behind it are 2.42 m and −3.3717 ms$^{-1}$. Boundary conditions adopt ghost cells at the conduct inlet and outlet. Before the two filling-bores collide with each other, the analytical result at $t_0$ is

$$
\begin{aligned}
h(x) &= \begin{cases} 3.167 \text{ m} & x < 10.067t_0 \\ 0.6 \text{ m} & 10.067t_0 < x < 200 - 8.429t_0 \\ 2.42 \text{ m} & 200 - 8.429t_0 < x \end{cases} \\
u(x) &= \begin{cases} 4.0334 \text{ ms}^{-1} & x < 10.067t_0 \\ 0 \text{ ms}^{-1} & 10.067t_0 < x < 200 - 8.429t_0 \\ -3.3717 \text{ ms}^{-1} & 200 - 8.429t_0 < x \end{cases}
\end{aligned}
\tag{38}
$$

In P_HLL, we choose $P_a$ = 5 and $P_b$ = 0.8, optionally. In M_HLL, we choose $K_a$ = 1.4 and $NS$ = 5 as suggested by Malekpour and Karney. The computational cell is 1 m, time step is 0.0008 s and the Courant number is 0.8. The simulation results in the two tests at $t$ = 6 s are drawn in Figure 16. In this

paper, an error indicator based on the $L_2$-norm [34] is used to evaluate the accuracy of P_HLL and M_HLL. In the following equation, $\varphi_i$ stands for the simulation result at cell $i$, while $\varphi_{\text{ref}}$ stands for the analytical result.

$$L_2 = \left( \frac{1}{N} \sum_{i=1}^{N} (\varphi_i - \varphi_{\text{ref}})^2 \right)^{0.5} \tag{39}$$

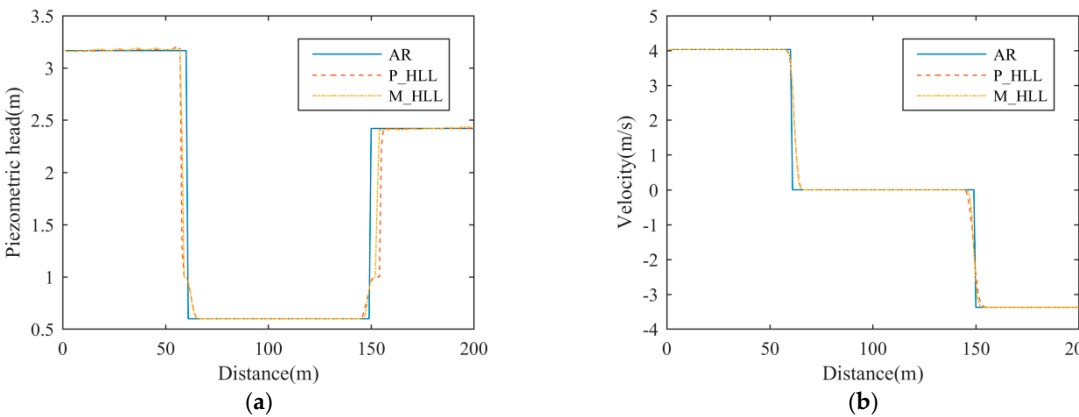

(**a**)　　　　　　　　　　　　　　(**b**)

**Figure 16.** Comparison of the results simulated by P_HLL ($P_a = 5$ and $P_b = 0.8$), M_HLL ($K_a = 1.4$ and $NS = 5$) and the analytical result (AR): (**a**) piezometric head; (**b**) flow velocity.

The $L_2$ in the piezometric head of P_HLL and M_HLL are 0.2963 and 0.2913, respectively; the $L_2$ in the velocity of P_HLL and M_HLL are 0.2879 and 0.2873, respectively. In P_HLL, the spreading length of the right filling-bore is slightly longer than that in M_HLL. This denotes that P_HLL is more diffusive than M_HLL in there. At the same time, P_HLL has eliminated some minor numerical oscillations while M_HLL does not. Both solvers are very robust and stable.

*5.2. Negative Pressure Flow*

In this test, P_HLL and M_HLL are adopted to simulate a water hammer phenomenon in a 600 m-long circular pipe with a 0.5 m diameter and acoustic wave speed is 1200 ms$^{-1}$. The pipe is horizontal and frictionless; a steady flow rate of 0.477 m$^3$s$^{-1}$ is initially in it. It connects to a reservoir at the downstream end, and water depth inside it is 45 m; see Figure 17.

At $t = 0$ s, the inflow rate is decreased to 0.4 m$^3$s$^{-1}$, which triggers a water hammer phenomenon; the water hammer pressure is 48.05 m according to Kerger et al. [26]. In P_HLL, the values of $P_a$ and $P_b$ are 100 and 0.8. In M_HLL, the values of $K_a$ and $NS$ are 1.2 and 12, as suggested by Malekpour and Karney. The computational cell is 1.2 m, the time-step is 0.0008 s and Courant number is 0.8. A ghost cell is set at the upstream boundary, and flow rate in it is constantly 0.4 m$^3$s$^{-1}$, while the piezometric head adopts the transmissive condition. Another ghost cell is set at the downstream boundary in which the piezometric head is constantly 45 m and the flow rate adopts the transmissive condition.

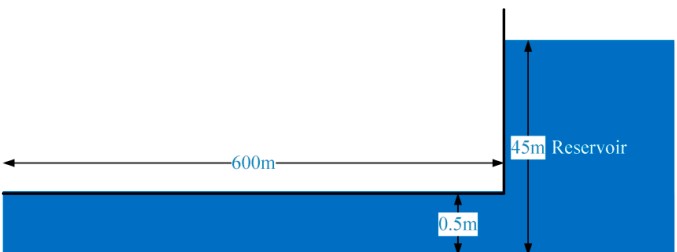

**Figure 17.** A sketch of the conduct–reservoir model.

The $L_2$ indicator is adopted to evaluate the accuracy of the two solvers; it is defined as

$$L_2 = \left( \frac{1}{N} \sum_{i=1}^{N} (\varphi_i - \varphi_{\text{ref}})^2 \right)^{0.5} \tag{40}$$

where $N_t$ is the number of time step, $\varphi_i$ is the simulation result at the midpoint of the pipe, and $\varphi_{\text{ref}}$ is the analytical result, which is given as

$$h_i = \begin{cases} 45 \text{ m} & n < i\Delta t < n + 0.25 \\ -3.05 \text{ m} & n + 0.25 < i\Delta t < n + 0.75 \\ 45 \text{ m} & n + 0.75 < i\Delta t < n + 1.25 \\ 93.05 \text{ m} & n + 1.25 < i\Delta t < n + 1.75 \\ 45 \text{ m} & n + 1.75 < i\Delta t < n + 2 \end{cases} , n = 0, 1, 2, 3 \ldots$$

$$u_i = \begin{cases} 2.4293 \text{ ms}^{-1} & n < i\Delta t < n + 0.25 \\ 2.0377 \text{ ms}^{-1} & n + 0.25 < i\Delta t < n + 0.75 \\ 1.6461 \text{ ms}^{-1} & n + 0.75 < i\Delta t < n + 1.25 \\ 2.0377 \text{ ms}^{-1} & n + 1.25 < i\Delta t < n + 1.75 \\ 2.4293 \text{ ms}^{-1} & n + 1.75 < i\Delta t < n + 2 \end{cases} , n = 0, 1, 2, 3 \ldots \tag{41}$$

The history of the flow states at the pipe midpoint in the simulation and analytical results are drawn in Figure 18. Both solvers nicely capture the reflection of the water-hammer wave in the pipe. The $L_2$ in the piezometric head of P_HLL and M_HLL are 6.3965 and 6.3970, respectively, while $L_2$ in the velocity of P_HLL and M_HLL are 0.1332 and 0.1333, respectively.

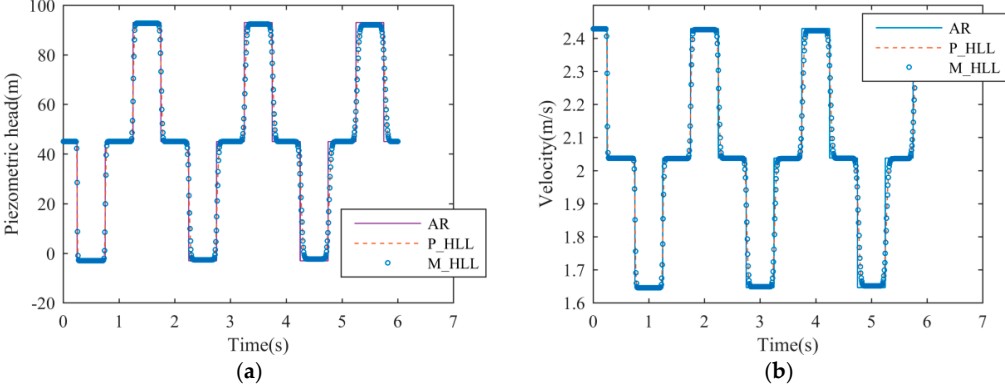

**Figure 18.** Locus of the flow states at the midpoint of the pipe simulated by P_HLL ($P_a = 100$ and $P_b = 0.8$), M_HLL ($K_a = 1.2$ and $NS = 12$) and the analytical result (AR): (**a**) piezometric head; (**b**) flow velocity.

Interestingly, in this test, the P_HLL solver can obtain almost the same result with an arbitrary value of $P_a$; for example, the simulation result using $P_a = 1$ is drawn in Figure 19.

In Section 4, we have proven that under the framework of PSM, any non-negative value of $P_a$ will produce a wave speed that is close to the acoustic wave speed, provided that the cell is under pressurized flow condition; see Figure 12 for detail. In this test, all the computational cells are under a pressurized flow condition, which makes the simulation result of $P_a = 100$ and $P_a = 1$ almost the same. In the flow regime transition simulation, $P_a H$ must be larger than the highest piezometric head.

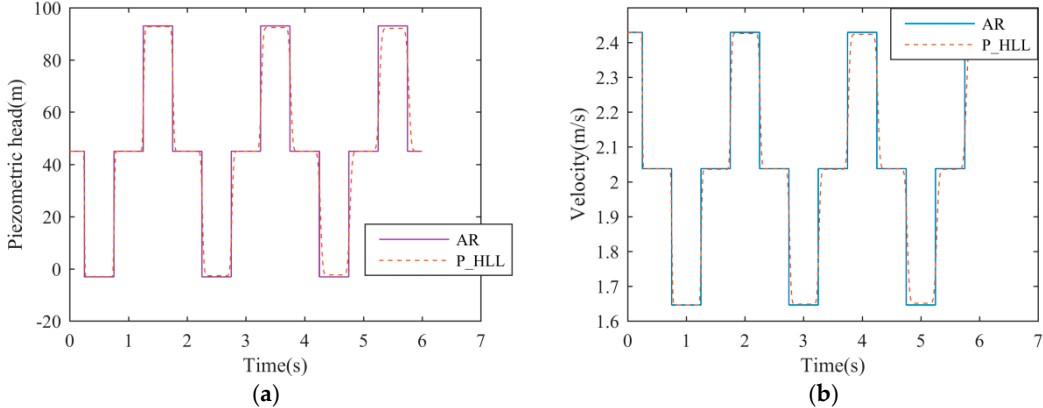

**Figure 19.** Locus of the flow states at the midpoint of pipe simulated by P_HLL ($P_a = 1$ and $P_b = 0.8$) and the analytical result (AR): (**a**) piezometric head; (**b**) flow velocity.

### 5.3. Vasconcelos's Experiment

This experiment was designed by Vasconcelos et al. [35] to study the filling process in a realistic stormwater storage tunnel. It is a 14.6 m-long horizontal tunnel with circular cross-sections of 9.4 cm in diameter; initially, stagnant water of 7.3 cm depth is established in the tunnel. A fill box with a 25 cm × 25 cm bottom and 31 cm height connects to the tunnel inlet. A surge tank with a constant diameter of 19 cm connects to the tunnel outlet. The experiment starts by constantly supplying 3.1 Ls$^{-1}$ water into the fill box, and when water level inside the fill box reaches its top, water is simply spilled away. A gate is installed at the tunnel outlet; its opening is smaller than the initial water depth. When the filling bore collides with the gate, it triggers a water-hammer phenomenon. A ventilation tower is fixed upstream of the gate so that no air is trapped in the tunnel. The experiment setup is drawn in Figure 20.

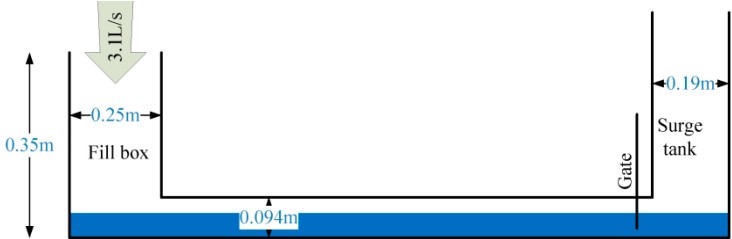

**Figure 20.** A sketch of the experimental setup.

The manning coefficient is 0.016 m$^{1/6}$, acoustic wave speed is 300 ms$^{-1}$ and head loss coefficient associated with the partially opened gate is 12, as suggested by Malekpour and Karney. In P_HLL, the values of $P_a$ and $P_b$ are 5 and 0.8. In M_HLL, the values of $K_a$ and $NS$ are 1.2 and 12. The computational cell is 0.1 m, and the time-step is set for a Courant number of 0.8. At the upstream end, the three unknowns are the discharge, the wetted area at the tunnel inlet and the water level in the fill box. At the downstream end, the three unknowns are discharge, the wetted area at tunnel outlet and the water level in the surge tank. The continuity, energy and characteristic equations are applied to obtain the three unknowns at each boundary [36]. The loci of flow states at $x = 9.9$ m in the simulation results of P_HLL and M_HLL are shown in Figures 21 and 22.

The simulation results are in good agreement with the experimental data, and the two solvers have correctly computed the arrival time of the filling-bore front at $x = 9.9$ m. P_HLL has computed a piezometric head peak slightly larger than M_HLL. Most importantly, P_HLL and M_HLL do not smear the physical pressure oscillations.

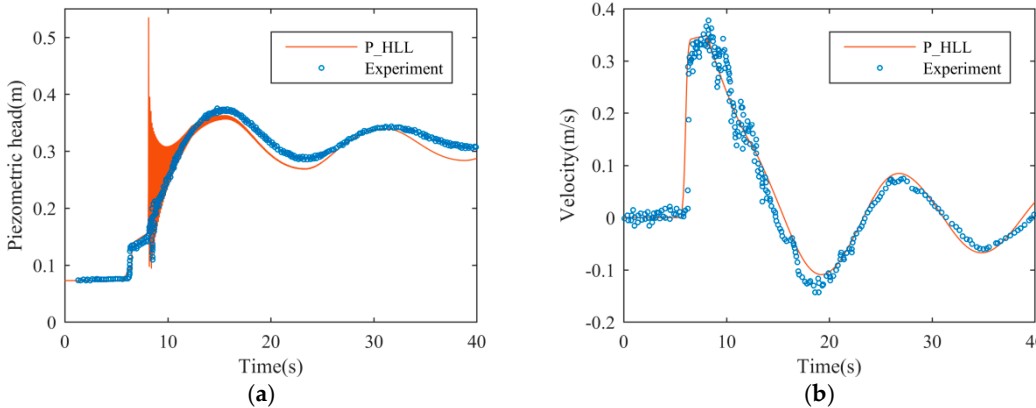

**Figure 21.** Locus of flow states at $x = 9.9$ m simulated by P_HLL ($P_a = 5$ and $P_b = 0.8$) and experimental data: (**a**) piezometric head; (**b**) flow velocity.

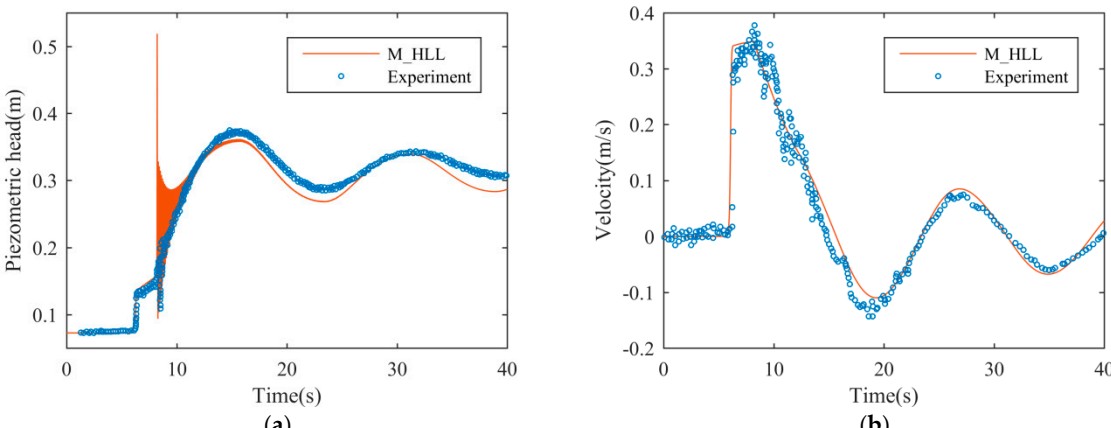

**Figure 22.** Locus of flow states at $x = 9.9$ m simulated by M_HLL ($K_a = 1.2$ and $NS = 12$) and experimental data: (**a**) piezometric head; (**b**) flow velocity.

## 5.4. Aureli's Experiment

This experiment is conducted on a 12.12 m-long pipe with a 19.2 cm-diameter and 4 mm-wall thickness [25]. We used $x$ to present the longitudinal distance to the pipe inlet. There is a sharp turn at about $x = 7$ m; the downward part has a slope of about 8.4% and the upward part has a slope of $-27.7\%$. Six piezometric transducers were installed in the bottom at $x = 1$ m, 3 m, 4.5 m, 6.8 m, 7.06 m and 8.52 m. A sluice gate was installed at approximately $x = 5$ m; it is closed at the initial conditions. Stagnant water with a 22.5 cm piezometric head at transducer 1 was established behind the sluice gate; the rest of the pipe was empty. As the experiment began, the gate was lifted within 0.2 s, setting flush into the pipe. The pipe inlet was blocked so that no water flows through it, while the outlet was totally open. The experimental setup is shown in Figure 23.

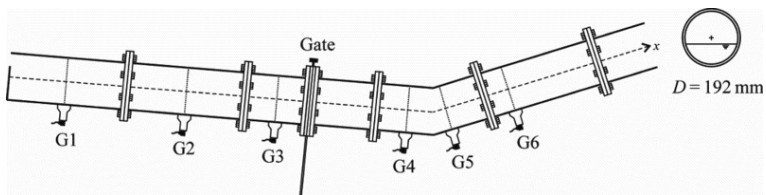

**Figure 23.** A sketch of the experimental setup, drawn by Aureli et al. [25].

In P_HLL, the values of $P_a$ and $P_b$ are 4 and 0.7; in M_HLL, the values of $K_a$ and $NS$ are 1.4 and 5. The computational cell is 0.04 m, acoustic wave speed is 200 ms$^{-1}$ and time-step is set for a Courant

number of 0.8. A reflective boundary condition was set at the upstream end, while a transmissive boundary condition was set at the downstream end. At the wet/dry interface, the estimates of wave speed followed Leon et al. [27]. The loci of the flow states at $x = 6.8$ m simulated by P_HLL and M_HLL are drawn in Figures 24 and 25.

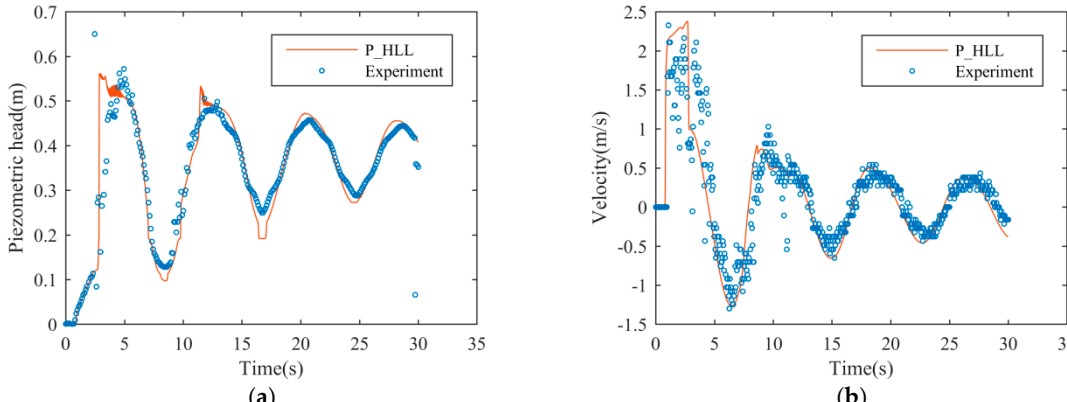

**Figure 24.** Locus of flow states at $x = 6.8$ m simulated by P_HLL ($P_a = 4$ and $P_b = 0.7$) and experimental data: (**a**) piezometric head; (**b**) flow velocity.

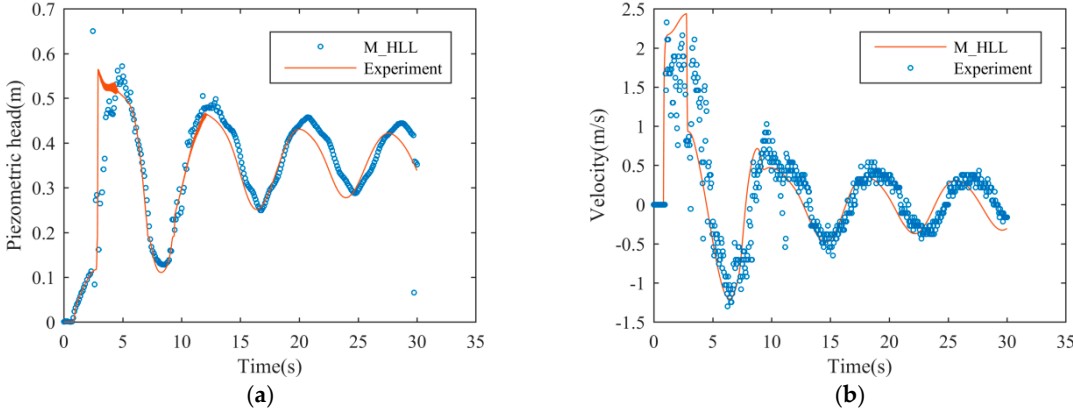

**Figure 25.** Locus of flow states at $x = 6.8$ m simulated by M_HLL ($K_a = 1.4$ and $NS = 5$) and experimental data: (**a**) piezometric head; (**b**) flow velocity.

The simulation results of P_HLL and M_HLL show certain discrepancies: M_HLL overestimates the wave speed. It is because P_HLL adds numerical viscosity only if the depth is higher than the threshold value of $P_b H$, while M_HLL adds numerical viscosity at any free-surface cells. Nonetheless, the simulation results of the two solvers are in good agreement with the experimental data.

## 6. Conclusions

Numerical oscillation is a critical problem in transient mixed flow simulations. These numerical oscillations arise from the ambiguity about the location of the filling-bore within one computational cell, which cannot be captured even with high-order finite volume methods. First order finite volume methods have failed to suppress them while capturing the filling-bore front.

Four oscillation-suppressing methods were tested on a benchmark model, with three of them failing to get a satisfactory result under a high acoustic wave speed. The key is to admit numerical viscosity before the flow regime transition occurs. Numerical viscosity can be added by artificially increasing the magnitude of the wave speed in the HLL solver. Following this idea, this paper presents a P_HLL solver that has two parameters: $P_a$ and $P_b$. $P_a$ needs to be larger than the highest piezometric head while $P_b$ needs to be between 0.7 and 0.9. This solver adds numerical viscosity when the water

depth is above $P_bH$ so that a smooth transition between the free-surface and pressurized flows can be achieved. The amount of numerical viscosity increases with $P_a$, while a large $P_a$ does not smear the simulation result significantly. Therefore, one can always start by adopting a $P_a$ that is large enough under realistic applications, for example 10.

The P_HLL solver is tested in several numerical tests, in which a good agreement between the simulation results and analytical results or experimental data is found. In the test where the wet/dry interface is presented, the P_HLL solver achieves a more accurate result than M_HLL. Meanwhile, P_HLL solver has sufficiently suppressed the numerical oscillations, and accurately captured the propagation of the filling-bore. Compared to the M_HLL solver proposed by Malekpour and Karney, the P_HLL is more convenient to use as the parameters in this solver are easier to determine.

The result in this paper can provide useful information for readers to design an oscillation-suppressing method. It provides an alternative method to the M_HLL solver, which may be used in the parallelization of computation.

This method is proposed under the framework of PSM; it is limited to the simulation of flow regime transition. Besides, since the air pressure is not counted, this method cannot be applied to simulate flow regime transition where air pockets are present, which is out of the scope of this paper.

**Author Contributions:** Conceptualization, Z.M., G.G., Z.Y.; resources, Z.M., G.G., Z.Y.; writing—original draft, Z.M.; writing—review and editing, Z.M. All authors have read and agreed to the published version of the manuscript.

**Funding:** The authors acknowledge the support of the NSFC grant 51979202 and NSFC grant 51879199.

**Conflicts of Interest:** The authors declare no conflict of interest.

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
