# Peer review of "Suppress Numerical Oscillations in Transient Mixed Flow Simulations with a Modified HLL Solver"

_water, doi:10.3390/w12051245_

Round 1

Reviewer 1 Report

I think the paper in the present form can be published in the journal. A good piece of work with a very good presentation. Congratulations to the authors!   

Author Response

We appreciate the time and effort spent on reviewing this paper, and we thank the reviewer for the consent of publication on water, this encourages us to refine this paper and continue our study.

Reviewer 2 Report

The manuscript proposes a new method for suppressing numerical oscillations during transient flow simulations. The manuscript is well structured and the presented approach is well supported by various tests. However, the paper needs some refinement before publication.

1. It would be nice to have visual representation for each test: the schematic of the setup with all the dimensions mentioned in text. It would help a reader to digest the information.

2. Names like test1 and test2 in figure 9 (as well as in text itself) are even less descriptive than hybrid one and hybrid two. I suggest the authors to come up with better and mode distinguishable names for all the different approached presented.

3. Why another modified HLL solver was not used on the last 2 test cases? It would be more consistent to have it on all 4 tests.

4. For the readers who are not familiar with the problem, it would be not clear how bad are the oscillations without suppression. Maybe it is possible to show this case for the benchmark problem as well? Or mention that it is not possible.

5. The authors gave the range for Pb parameter between 0.7 and 0.9, but in test 4 they use 0.6. Why? Are there any guidance why this or the other value should be chosen for Pb except for it to be less that one?

Also, it says Pa = 0.8 and Pb = 100 in test 2. (line 234) I suppose it should be the other way around.

6. Are there limits for this approach, when it cannot be applied?

7. The text was not proofread before submission – there are typos, missing words and extremely long, unreadable sentences. It was unreasonably hard to get through all the text. Below, I pointed out some of the typos, but then I gave up, and I encourage the authors to go through the manuscript text in detail.

Line 9. “is an a crucial phenomenon”

Line 15-17. “Then a new oscillation-suppressing approach is proposed to admit numerical viscosity when the water surface is at proximity of conduct roof which has first order accuracy. This approach adds numerical viscosity when water surface is at the proximity of conduct roof.” Essentially the same sentence is repeated twice.

Line 34. “it is meaningful” I suggest to use another synonym for more precise English meaning, for example “important” or “essential”.

Line 41. “The gradient of piezometric head at the interface between two flow regimes are is very steep”

Line 45-49 “These numerical oscillations are very similar to the “post-shock oscillations” in gas dynamics which have been studied by many researchers [20-22]. It is pointed out that the numerical oscillations in transient mixed flow simulations have the same sources as the “post-shock oscillations” in gas dynamics [23].” Again, essentially the same sentence repeated twice.

Line 65, 68, 69 Numbering of the sections is wrong. Benchmark problem is introduced in section 3 together with oscillation-suppressing methods, which means that section 2 is not mentioned in introduction at all. New method is introduced in section 4 and validation is done in section 5.

Line 102 “inside a this conduct” ??

Reviewer 3 Report

Please you can find the review of the manuscript in the attached pdf file.

Author Response

Please you can see response to your comments in the attachment.

Round 2

Reviewer 2 Report

I would like to thank authors for their great work on improvement of the manuscript. It was a pleasure to read. And I am satisfied with authors response to my comments. I recommend this version for publication.

Minor remarks:

line 10:  "The simulation of" - I believe it should be "During simulation of"

line 35 and line 325: I don't see the reason to use the word "while"

line 37: "appear" should be "appearance"

line 87: "insider"should be "inside"

line 146: "The reason is that," the comma is not needed

line 151-152: the second part of the sentence doesn't have a verb.

Author Response

The authors want to thank the reviewer for the reviewing with great carefulness and patience, and we appreciate the support to publication of our work. The remarks on expressing have been checked carefully and corrected in the revised manuscript. 
line 10:  "The simulation of" - I believe it should be "During simulation of"

Response: It has been corrected in line 10.

line 35 and line 325: I don't see the reason to use the word "while"

Response: The word "while" has been removed, see line 35 and 324.

line 37: "appear" should be "appearance"

Response: The word "appear" has been removed in line 37.

line 87: "insider"should be "inside"

Response: It has been corrected in line 87.

line 146: "The reason is that," the comma is not needed

Response: The comma has been removed in line 145.

line 151-152: the second part of the sentence doesn't have a verb.

Response: The verb "increases" has been added in line 151.

Thank you for your instructive comments!

Reviewer 3 Report

The revised version of the manuscript has be significantly extended and improved with respect to the previous one. Most of the points I raised in the first review were addresed. I support the publication of the manuscript in the present form. A few minor changes are suggested in the following.

- I remark that the first sentence of the Introduction (lines 24-25) should be removed, also because it is not correct (free-surface and pressurised conditions are characteristics of currents, whereas water can also be at solid and gasseous states).

- I suggest to replace the symbol "&" with "and" in equation (27).

- Line 190: the colon should be replaced by full stop.

- Line 196: "Therefore, P_HLL preserves mass and momentum." Anyway, this conclusion is still quite heuristic because convergence is not rigorously proved.

- Line 204: the colon should be replaced by full stop.

- Line 246: "After all, both solvers are very robust and stable" -> "Both solvers are robust and stable".

Author Response

The authors want to thank the reviewer for the consent to the publication of this paper. We appreciate the effort that has been spent during the reviewing. The reviewer's expertise and instructive comments help us to improve our work. The response to your comments are listed below.

  • I remark that the first sentence of the Introduction (lines 24-25) should be removed, also because it is not correct (free-surface and pressurised conditions are characteristics of currents, whereas water can also be at solid and gasseous states).
  • Response: The first sentence has bee changed to "In water conveyance systems, water flows under free-surface flow condition or pressurized flow condition."
  • - I suggest to replace the symbol "&" with "and" in equation (27).
  • Response: The symbol has been changed in equation 27.
  • - Line 190: the colon should be replaced by full stop.
  • Response: The colon has been replaced by full stop.
  • - Line 196: "Therefore, P_HLL preserves mass and momentum." Anyway, this conclusion is still quite heuristic because convergence is not rigorously proved.
  • Response: A new figure depicting the history of flow states inside cell i+1 is added as Fig 10(b). It can been seen that the simulation result of P_HLL converges to the final steady state in a smooth behavior.
  • - Line 204: the colon should be replaced by full stop.
  • Response: The colon has been replaced by full stop.
  • - Line 246: "After all, both solvers are very robust and stable" -> "Both solvers are robust and stable".
  • Response: The sentence has been modified.

Thank you very much!